# Monoclonal Gammopathy of Thrombotic Significance

**DOI:** 10.3390/cancers15020480

**Published:** 2023-01-12

**Authors:** Vasiliki Gkalea, Despina Fotiou, Meletios Athanasios Dimopoulos, Efstathios Kastritis

**Affiliations:** Department of Clinical Therapeutics, School of Medicine, National and Kapodistrian University of Athens, 15772 Athens, Greece

**Keywords:** monoclonal gammopathy of undetermined significance, monoclonal gammopathy of clinical significance, thrombosis, monoclonal immunoglobulin, coagulation biomarkers, cryoglobulins, cryofibrinogenemia, cryo-crystaloglobulinemia, autoantibodies

## Abstract

**Simple Summary:**

It is being increasingly recognized that patients with monoclonal gammopathy (MG) who do not meet criteria for malignant disease and treatment initiation may have other manifestations that are linked to the unique properties of the monoclonal protein that are clinically significant and could provide a rationale for treatment. It has been recognized that patients with MG are at higher risk for thrombosis and a number of rare entities that include venous, arterial and micro-thrombotic events have been described. The hemostatic profile of patients with MG has not been ascertained yet and the major challenge is to identify patients with MG at high risk for thrombotic events and apply appropriate preventive measures. We propose a new umbrella term, monoclonal gammopathy of thrombotic significance, that creates a link between the M-protein and thrombotic events, posing the question of whether treatment initiation and/or long term anticoagulation is indicated for these, otherwise asymptomatic patients.

**Abstract:**

The current review provides an overview of the thrombotic risk observed in patients with MG who do not otherwise require treatment. We discuss clinical and biomarker studies that highlight the heterogenous hemostatic profile observed in these patients and how knowledge has evolved over the past 20 years. Biomarker studies suggest shared biologic features between multiple myeloma and monoclonal gammopathy of undetermined significance (MGUS), which involves both hypercoagulability and platelet activation. Hemostatic abnormalities identified in MGUS patients cannot be translated into clinical practice as they lack correlation to clinical events. The prothrombotic phenotype of MGUS patients has not been ascertained yet, but novel data on coagulation markers are promising. We also review rare conditions associated with the thrombogenic properties of the monoclonal protein that predispose to arterial, venous or microthrombotic events and demonstrate that the M-protein can be linked to clinically significant thrombotic events. Cryoglobulinemia, cryofibrinogenemia, cryo-crystaloglobulinemia and MG-related antiphospholipid syndrome are reviewed. We propose the new umbrella term “monoclonal gammopathy of thrombotic significance” (MGTS) to refer to significant, recurrent thrombotic events in patients with MGUS that provide a rationale for targeting the underlying plasma cell clone. Identifying MGUS patients at high risk for thrombotic events is currently a challenge.

## 1. Introduction

Monoclonal gammopathy (MG) refers to plasma cell dyscrasias (PCD) and lymphoproliferative disorders (LPD) characterized by the presence of a monoclonal immunoglobulin (M-protein) or monoclonal light chain produced by the clonal cell population [1]. Clinical expression of MG varies widely from a benign form to the full-blown multiple myeloma (MM). The benign form of MG, defined as monoclonal gammopathy of undetermined significance (MGUS), is present in 3.2% of Caucasians aged 50 years [2]. Based on the recent IstopMM trial, the prevalence is 3.9% in ≥40 years old and 5% in the ≥50 years old population [3]. MGUS carries an annual risk of 1% of progression to MM, while almost all cases of MM are preceded by MGUS [4]. MGUS is most often diagnosed as an incidental finding, but can be also discovered during investigations for other non-malignant comorbid medical conditions.

Naïve untreated MM patients have an increased inherent risk for venous thrombosis which further increases during anti-MM treatment [5]. To date, several clinical studies suggest that patients with MGUS may as well carry a higher venous thromboembolism (VTE) risk compared to the general population [6]. Nevertheless, the debate remains open since recently emerged data point out that the increased thrombotic risk could be related to accompanying medical conditions in patients with MGUS, rather than to a blood-borne hypercoagulability, indicating selection bias in previous studies [7].

On the other hand, biomarker studies suggest that there might be some shared biologic features between MM and MGUS responsible for the increased thrombotic risk, involving platelet activation and hypercoagulability [8,9]. Nevertheless, hemostatic abnormalities identified in MGUS patients lack correlation to clinical events and cannot be translated to clinical practice. Moreover, the M-protein has known thrombogenic properties that have a proven causative relationship with thrombotic events, and have been described in otherwise asymptomatic MGUS patients or in patients with MG not fulfilling treatment criteria [10,11]. Currently, prophylaxis and treatment of thrombotic events in MGUS patients and in untreated MG patients follow the same principles with the general population [12,13].

The complex pathophysiologic interactions between M-protein, platelets, coagulation factors and endothelium can also produce hemostatic abnormalities that increase the bleeding risk [14]. It is beyond the scope of this review to discuss MG-related bleeding disorders such as acquired von Willebrand syndrome, FX deficiency and platelet function disorders. However, the notion that M-protein can cause both thrombotic and hemorrhagic tendencies emphasizes further its unique physicochemical properties.

This review will address the issue of the thrombotic risk in MG patients not requiring treatment, including MGUS patients. Data from both clinical and biomarkers studies will be considered. It aims to highlight the heterogeneity of the design of the clinical studies and summarize the knowledge evolution over the past 20 years. In addition, the understanding of the hemostatic profile in MGUS patients remains incomplete. Biomarker studies do not ascertain a prothrombotic phenotype in MGUS patients in a consistent manner and are largely undermined by the small numbers of patients included. However, data on novel coagulation markers are promising and may shed some light on the complex relationship between hemostasis and MG. Finally, rare conditions associated with the thrombogenic properties of the monoclonal immunoglobulin sometimes complicate the clinical picture and management of patients with MG who do not otherwise require treatment. The pathogenesis, clinical presentation and treatment of these rare entities will be discussed.

## 2. Literature Review

### 2.1. Clinical Trial Data

Clinical studies on the thrombotic risk associated with MGUS confer contradictory results. They are mainly single-center, hospital or population-based, and include MGUS patients identified either through screening or during a medical evaluation for related issues (Table 1). Outcomes concern mainly the risk of venous thromboembolism, whereas data regarding the risk of arterial events are scarce.

The first data regarding the risk of VTE in MGUS patients comes from two single-center case series published in 2004. Both studies demonstrated a high VTE risk in MGUS patients compared with the general population. A 7.5% VTE incidence (median time of 4 months from diagnosis to VTE) was observed among 174 MGUS patients identified from a retrospective review of the medical records over a ten-year period [15]. The VTE incidence in MM patients was 10% and median time of occurrence was 8.5 months from diagnosis. Classical VTE risk factors, low albumin levels and high leucocyte count, were independently associated with VTE risk [2]. No correlation between VTE and disease progression was observed. Similarly, a prospective evaluation of VTE risk in a cohort study of 310 newly diagnosed MGUS patients with a median follow up of 44 months reported a 6.1% VTE rate, with a median time to event occurrence of 23 months following diagnosis. M-protein levels > 16 g/L, and disease progression to myeloma or amyloidosis were associated with increased VTE risk [16]. Both studies were lacking control groups and VTE incidence in MGUS patients was compared to that observed in the general population which approximates 1/1000/year [24].

Kristinsson and colleagues studied, retrospectively, 2374 MGUS cases identified among more than 4 million adult male military veterans admitted at least once to US Veterans Affairs (VA) hospitals [17]. During 17 years of follow-up, a 3.3-fold increase in deep venous thrombosis (DVT) risk was observed in MGUS patients and a 9.2-fold increase in MM patients compared to the entire VA population. DVT risk was stable over time in MGUS cases and highest during the first year after diagnosis among MM patients. Only one MGUS case had a diagnosis of DVT prior to disease progression to symptomatic MM. The same group confirmed their results using population-based data from Sweden [18]. The risk of venous and arterial thrombosis was assessed in 18,627 MM and 5326 MGUS patients diagnosed from 1958 to 2006 and compared to age- and gender matched controls. The risk of VTE and arterial thrombosis was highest during the first year following MGUS diagnosis with Hazard Ratios (HR) of 3.4 and 1.7, respectively. During the 10-year follow-up the risk was 2.1 times higher and 1.3 times higher for VTE and arterial thrombosis, respectively. The HR for VTE risk in MM patients was 7.5 and 4.1 at 1 and 10 years, respectively, and 1.9 and 1.5 for arterial thrombosis, respectively. Patients with IgG and IgA (but not IgM) MGUS had a four-fold increased risk of venous thrombosis. This study demonstrated, for the first time, that MGUS patients with thrombosis had inferior survival compared to MGUS patients without thrombosis. Patients with thrombosis had no excess risk of MM or Waldenström’s macroglobulinemia.

In another MGUS cohort from a single Veterans Affairs healthcare system, the VTE rate (2.2 per 1000 person-years) was not significantly higher compared to a control population negative for the presence of M-protein (1.4 per 1000 person-years) [19]. Most VTE events occurred within 4 months of the diagnosis of MGUS. Patients with MGUS and VTE were more likely to have IgA heavy chain or lambda light chain. Albumin level and personal history of VTE remained significant predictors of VTE. No patient progressed to myeloma during 413 total person-years of follow-up. A diagnosis of underlying malignancy was present in 19% of MGUS cases and 16% of controls. Muslimani et al., reviewed, retrospectively, the medical records of 112 MGUS patients from a single center during a 3-year period [20]. Patients were divided into two groups according to the risk for progression to myeloma. During a median follow-up of 867 days, 9 patients (8.0%) with MGUS experienced VTE. A non-significant trend towards increased VTE risk was noticed in the higher-risk compared to the lower-risk group (6.8 vs. 11.8%). However, numbers in each group are too small to allow for safe conclusions. Bida et al. [21] studied the association of MGUS (n = 605) with all diseases in a population-based cohort of 17,398 patients, all of whom were uniformly tested for the presence or absence of MGUS. Superficial thrombophlebitis was a previously unreported associated condition.

More recently, a large population-based cohort study, using the Danish Central Population Registry, confirmed that MGUS is a risk factor for VTE, and VTE is a marker for increased mortality [22]. Fifty VTE events among 1610 MGUS patients were identified (incidence rate of 4.0 VTEs/1000 person-years), with 12,594 person-years (PY) of follow-up. The VTE incidence was increased by 37% compared to a cohort of 16,100 subjects with no prior VTE diagnosis, matched with MGUS patients for age, sex, and comorbidities. One out of the 50 MGUS patients with VTE progressed to myeloma. The adjusted mortality rate ratio for MGUS patients with VTE compared to MGUS patients without VTE was 1.94. In contrast, no increase in thrombotic events was reported in a multicenter, retrospective study, in a cohort of 1491 MGUS patients [23]. In total, in 1238 patients with a follow up >12 months, 33 patients (2.7%) had a venous or arterial thrombotic event (median follow up of 3.3 years). No thrombotic events were observed in MGUS patients who progressed to MM or developed other malignancies. A threefold increased risk for VTE was found among patients with M-protein values >16 g/L. More recently, a population-based study with an 8-year follow-up from the longitudinal cohort of the AGES-Reykjavik Study (5764 older-age individuals) identified 299 patients with MGUS. No increased risk of arterial or venous thrombosis was shown. A history of thrombosis was more common in LC-MGUS patients, which might be linked to the older age of LC-MGUS individuals in this cohort. The results from this screened study contradict previous findings from clinically established cohorts [7]. Of note, this was the first study to investigate the risk of thrombosis in LC-MGUS.

Another large population-based screening study for MGUS, iStopMM, included 54% (n = 80,759) of the Icelandic population above 40 years of age and compared 224 individuals with incidentally diagnosed MGUS and 3076 with screened MGUS. Individuals with incidentally diagnosed MGUS also had a higher number of underlying comorbidities, but the incidence of thrombotic complications was not higher [25]. Finally, a large screening study compared mass-spectrometry to conventional methods for detecting serum monoclonal proteins in high-risk individuals (>50 years old). All MGUS patients had a higher prevalence of associated clinical outcomes such as decreased overall survival and increased likelihood of comorbidities, including myocardial infarction (odds ratio 1.60 for MGUS with Mpeak < 0.2 g/L and 1.39 for mass-spectrometry diagnosed MGUS) [26].

### 2.2. Biomarker Studies (See Table 2 and Table 3)

#### 2.2.1. Evidence of In Vitro Hypercoagulability in MGUS Patients

Overall, MGUS patients are under-represented in the studies evaluating the hemostatic profile in PCDs. Very few studies have compared the coagulation and platelet activity profiles of patients with MM, MGUS and normal controls. Even though the presence of signs of hypercoagulability in MGUS patients is a common observation, the small number of patients included in the studies and the absence of correlation of in vitro evidence to thrombotic events discourages the application of current findings in clinical practice. So far, some groups have studied panels of biomarkers assessing specific pathways of hemostasis, while others have opted for a more global approach.

Markers of hypercoagulability were studied in a group of 25 MGUS patients among 198 patients with PCD (MM, systemic amyloidosis and WM) [27]. Antigen and activity levels of von Willebrand factor (vWf) were significantly increased in all PCD patients compared to healthy controls. In subgroup analysis, vWf antigen and activity levels and Factor VIII levels were increased in patients with MM, MGUS and systemic amyloidosis, but not with WM. The highest FVIII and fibrinogen levels were observed in AL patients. The frequency of hereditary thrombophilia, factor V Leiden mutation and the prothrombin 20210A gene variants, as well as protein S activity, were similar in patients with PCD and controls. The incidence of lupus anticoagulant varied between 5% and 33% (MGUS and systemic amyloidosis, respectively). None of the tested patients had anticardiolipin IgM antibodies, IgG antibodies were observed in a few MM and WM patients. In this cohort, three patients (two with MGUS and one with systemic amyloidosis) and fourteen patients (10%) with newly diagnosed MM who received anthracycline-based multiagent chemotherapy developed a thromboembolic complication.

In a small cohort of PCDs (MM, SM, MGUS) patients, there was a non-significant trend for higher factor VIII and vWf antigen levels in MGUS patients compared to normal controls [28]. D-dimer levels were three times higher in MM patients and two times higher in MGUS patients compared to controls. No associated increase in inflammatory markers was present. However, a very wide distribution of CRP values was noticed in the MM group. Moreover, D-dimer increase was not related to clinical evidence of VTE, but was attributed to a higher turnover of fibrin formation and fibrinolysis driven by the underlying malignancy. Importantly, there was no significant difference in thromboelastographic parameters between the three groups; there was a trend for a larger α angle and maximal amplitude in the MM group. Overall, excluding the increase in D-dimer levels, no other solid data in favor of hypercoagulability in MGUS patients were observed in this study.

On the contrary, Nielsen et al. showed increased thrombin generation (TG) and activity of procoagulant phospholipids (PPL) in both newly diagnosed MM and MGUS patients compared to healthy volunteers [9]. There was a trend for increased TG in MM compared to MGUS patients. Cell-free DNA (cfDNA) was increased in MM patients, reaching a five to tenfold higher concentration in some MM patients than the controls. In contrast, no difference was observed in MGUS patients. Even though cfDNA is not a specific marker of neutrophil extracellular traps (NETs) and of increased thrombotic risk in MM patients, it is a promising marker; higher levels are associated with worse progression-free survival (PFS) and overall survival (OS) in MM patients [29]. Indeed, cfDNA may reflect NETs formation subsequent to neutrophil activation or cell apoptosis and necrosis, and has been found to increase in physiological conditions [30] and various malignant [31,32] and non-malignant pathological processes [33]. Myeloma cells may directly induce NETs release through activation of PAD4 and interestingly, data from mice models show delayed MM progression with PAD4 inhibitors [34]. An increase in tissue factor (TF) activity was observed in MM patients compared to MGUS patients and controls. Plasma cells in MM patients release TF-rich microparticles, which could be responsible for the hypercoagulability observed in the MM microenvironment [35].

**Table 2 cancers-15-00480-t002:** Main studies in MGUS, MM, SMM patients reporting biomarkers studied in relation with thrombotic events. Monoclonal gammopathy of undetermined significance (MGUS), healthy controls (HC) and multiple myeloma patients (MM), AL: AL amyloidosis, WM: Waldenstrom’s macroglobulinemia. NS: non-significant, x: no data reported, vWF: von-Willebrand factor, TEG: thromboelastography, TG: thrombin generation, PPL; procoagulant phospholipid, MV TF: microvesicle tissue factor, cf: cell-free, PRP: platelet rich plasma, GPIV: glycoprotein VI levels, APLA: antiphospholipid lupus antibodies.

	Study Population	Results	Thrombotic Events	Strengths	Limitations
1	125 MGUS 145 MM 20 AL 8 WM 25 HCs Auwerda et al., 2007 [36]	No difference in Hereditary thrombophilia, APLA, aPTT, D-dimer Increased vWFa, FVIII, fibrinogen in MGUS, MM and AL vs. HCs Increased vWF Ag levels in all patients except MGUS	2 MGUS 1 AL 14 NDMM (on anthracycline-based Tx)	Case-control study Evaluation of multiple biomarkers and report of clinical events	Limited patients’ number to establish a relationship between coagulation abnormalities and VTE
2	8 MGUS patients 8 MM patients 8 HCs Crowely et al., 2015 [28]	Prolonged mean PT, higher mean FVIII, vWf, RCo levels and 3-fold increase of mean D-dimer levels (3-fold) in MM vs. HCs 2-fold higher median D-dimer levels in MGUS vs. HCs No differences in aPTT, fibrinogen and TEG parameters	Not reported	Case control study	Small number of pts per group. Mixture of mainly diagnosed and relapsed MM patients
3	36 NDMM 19 MGUS/SMM 34 HCs Nielsen et al., 2019 [9]	Elevated thrombin generation and PPLct activity in MGUS and MM vs. HCs Higher MV-TF activity in MM vs. MGUS and vs. HCs Increased levels of the cf-DNA in MM vs. HCs	No symptomatic VTE	Data on novel biomarkers not extensively studied in PCS Clinical data on antithrombotic treatment and VTE history available Case-control study	Small numbers Smaller age difference between the patient groups to HCs Cellular origin of TF-expressing microvesicles not studied cfDNA is not a specific marker for NETs- dependent hypercoagulability
4	10 MGUS, 3 SMM 7 MM on Tx 8 MM post Tx Egan et al., 2014 [37]	Decrease of platelet aggregation to all agonists in MM pts vs. MGUS except for ADP Decrease of P selectin expression to all agonists in MM vs. MGUS Soluble GPVI levels similar in MM vs. MGUS Platelet hyporeactivity not associated to paraprotein levels	Not reported	Differing profile of platelet activation in MGUS, active and post treatment MM patients Evaluation of different pathways of platelet activation	Small cohort No control group MM treatment is not specified Antithrombotic treatment is not mentioned
5	19 MGUS 12 MM 4 SMM 20 HCs Sullivan et al., 2021 [8]	At resting platelets, increased CD63 and PAC-1 expression and annexin V in MGUS vs. HCs Decrease of P-selectin expression upon stimulation with ADP. Trend towards reduced platelet responsiveness in pts compared to HCs	Not reported	Complete panel evaluating various pathways of platelet activation Previously unassessed markers of platelet activation (CD63 and PLAs)	SMM patients were included in the MM group
8	10 MGUS 10 SMM 4 MM 10 HCs Gibbins et al., 2018 [38]	No differences in platelet aggregation, secretion, and fibrinogen binding in response to multiple agonists in pts vs. HCs Similar levels of platelet receptors in all patients vs. HCs	Not reported	Use of washed platelets Evaluation of platelet function during disease progression Study of different pathways with various agonists	Small cohort

**Table 3 cancers-15-00480-t003:** Comparison of coagulation biomarkers between patients with Monoclonal gammopathy of undetermined significance (MGUS), healthy controls (HC) and multiple myeloma patients (MM). All comparisons are statistically significant (*p* < 0.05) unless otherwise stated, where NS: non-statistically significant differences between compared groups. Only data from studies thatinclude MGUS patients are shown in the table, x: no data reported, vWF: von-Willebrand factor, TEG: thromboelastography, TG: thrombin generation, PPL; procoagulant phospholipid, MV TF: microvesicle tissue factor, cf: cell-free, PRP: platelet rich plasma, GPIV: glycoprotein VI levels.

	MGUS vs. HCs	MM vs. HCs	MM vs. MGUS
Hereditary thrombophilia	NS [27]	NS [27]	x
vWf activity	Increased [27], NS [28]	Increased [27,28]	NS [27]
vWf antigen	NS [27]	Increased [27,28]	NS [27]
FVIII	Increased [27], NS [28]	Increased [27,28]	NS [27]
Fibrinogen	Increased [28], NS [28]	Increased [27], NS [28]	NS [27]
D-dimer	NS [28]	Increased [28], NS [27]	NS [27] Increased
TEG	NS [28]	NS [28]	x
TG	Increased [9]	Increased [9]	NS [9]
PPL activity	Increased [9]	Increased [9]	NS [9]
MV TF activity	NS [9]	Increased [9]	Increased [9]
cf DNA	NS [9]	Increased [9]	NS [9]
Platelet aggregation in PRP	NS [37]	NS [37]	Decreased (except in ADP) [37]
Platelet aggregation in washed platelets	NS [38]	NS [38]	x
P-selectin, CD63, Annexin V, PAC1 at resting	Increased [8]	NS [8]	x
P-selectin expression upon activation	Decreased [8]	NS [8]	Decreased [37]
Soluble GPVI levels	NS [38]	NS [8]	NS [37]
Circulating levels of PAC-1 and P-selectin	NS [37]	NS [37]	x

#### 2.2.2. Platelet Reactivity in MGUS Patients

To date, scarce data exist on the platelet activation status in MGUS patients. Platelets seem to play a central role in the pathogenesis MM-related thrombosis, considering the increased risk of arterial thrombotic events [39] and the lowering effect of aspirin on both arterial and venous thrombosis in patients receiving lenalidomide [40].

Patients with MM present with defective platelet function at diagnosis that normalizes after treatment and achievement of complete remission [41,42]. Decreased platelet reactivity has been associated with increased mortality and high risk of VTE in cancer patients [43]. Exhaustion of platelets related to continuous platelet activation has been proposed as a possible mechanism to explain this finding.

Monoclonal immunoglobulin-induced inhibition of platelet function [44,45] has also been demonstrated. Two findings are worth mentioning from early in vitro experiments. Firstly, the inhibition of platelet function induced by the addition of M-protein to platelet rich plasma was evident at relatively low concentrations of paraprotein [46]. Secondly, such an effect was difficult to reproduce with washed platelets, suggesting that plasma derived elements, such as M-protein alone or in the presence of other cofactors, may be needed to suppress platelet activation.

Egan et al. suggested that progression of MGUS to MM may occur in parallel with a decrease in platelet responsiveness [37]. Indeed, decreased platelet aggregation and P-selectin expression with multiple agonists were observed in MM compared to MGUS patients. Platelet aggregation upon ADP was not significantly different, suggesting that although alpha granule secretion is affected, the overall response of ADP activated pathway remains unaltered. Moreover, the decrease of the response to collagen was not associated with an increase in soluble GPVI levels; therefore, it could not be attributed to increased GPVI shedding. In addition, platelet hypo-reactivity was not associated with paraprotein levels. Of note, three patients with SMM were included in the MGUS cohort. Evidence regarding platelet function status in MGUS patients compared to healthy subjects was lacking in this study.

A more recent study provides evidence that platelet hypo-responsiveness in MGUS patients results from the ongoing process of platelet hyperreactivity at the resting state. Moreover, platelet hypo-responsiveness in MM correlated with advanced disease and poor prognosis [8]. Indeed, the expressions of CD63 (marker of dense platelet granule secretion), annexin V (surrogate marker for procoagulant phospholipid expression) and PAC-1 expression (indicator of the activation status of the fibrinogen receptor) were increased in resting platelets in MGUS patients, while in the SMM/MM group, there was a non-statistically significant increase. Upon stimulation with ADP or a TRAP-6 agonist, a trend towards reduced platelet responsiveness in patients compared to healthy controls was noted for both MGUS and MM. There was a negative correlation between resting levels of annexin V, CD63 and PAC-1 and P-selectin levels in response to agonists, favoring the conclusion that when platelets are more reactive at rest, they become less reactive upon activation. No differences were seen between patient groups. Of note, the proportion of patients with SMM/MM with an increase in three or more markers was greater than for patients with MGUS, suggesting that a subpopulation of patients with activated platelets exists in the MM patient group. P-selectin expression was the marker that was increased in the minority of MGUS and MM patients. No significant differences were observed in platelet–leucocyte aggregates (PLAs) and Ig expression compared to controls.

These results contradict the findings from the study of Gibbins et al. [38], who found that platelet functional responses from MGUS and SMM patients were unaltered compared to healthy controls. This study included a wide panel of platelet function assessments, evaluating the profile of multiple platelet activation pathways in 10 MGUS patients, 10 SMM, 4 MM and 10 healthy controls. Platelet aggregation, secretion, and fibrinogen binding, performed on platelet rich plasma and on washed platelets, in response to multiple agonists as well as levels of key platelet receptors measured by flow cytometry, were similar in all group patients as compared to controls.

#### 2.2.3. Hyperviscosity in MGUS Patients

The presence of a paraprotein may be associated with increased blood viscosity and subsequent slowing of blood flow through the vessels. The hemorheological alteration present in MM and MGUS patients has been attributed to the abnormality of plasma or serum viscosity in contrast to polycythemia vera, where increased blood viscosity is related to an increased mass of red blood cells (RBC) [47]. This is mainly driven by the type and the concentration of M-protein, but data from ex-vivo experiments suggest that RBC properties could play a role as well. IgM is pentameric and very large in size (970 kDA), and serum viscosity can increase with IgM levels as low as 3 g/dL, and IgM levels of 6 g/dL or higher are associated with rapid development of hyperviscosity [48]. An ex-vivo study comparing normal controls to 21 MGUS subjects observed a significant increase in whole-blood viscosity at high shear rate and in plasma viscosity at low shear rate, compared to normal controls, suggesting the presence of RBC abnormalities as well [49]. Nevertheless, to date, the only abnormality of the red cell membrane documented in MGUS subjects seems to refer to membrane proteins consisting of paroxysmal nocturnal hemoglobinuria-like (PNH-like) defects [50,51].

#### 2.2.4. The Role of Genetic Abnormalities in Thrombosis in MGUS Patients

At the genetic level, a translocation between chromosomes 11 and 14 that can be found in up to 20% of patients with MM has been linked to the development of VTE. Chakraborty et al. reported data in blood from a MM cohort suggesting that abnormal metaphase cytogenetics could have added value as predictors of VTE occurrence within 12 months of diagnosis in newly diagnosed myeloma patients [52,53]. These data need to be followed up and confirmed with further studies. Del(17p) and t(4;14) are associated with higher risk of progression of MGUS to MM, but no data exist linking the presence of chromosomal defects and VTE in MGUS patients. Towards this direction, besides the fact that iFISH might help to identify MGUS patients at higher risk of progression, it might also identify MGUS patients carrying higher VTE risk, but these findings need to be investigated further [54].

#### 2.2.5. The Role of the Microenvironment in MG-Related Thrombogenesis

Levels of coagulation-related factors in bone marrow (BM) CD45+ cells in patients with MGUS, SMM, and MM and healthy controls have been explored. Messenger RNA (mRNA) expression of tissue factor pathway inhibitor (TFPI) and thrombomodulin was significantly decreased in CD45+ cells in the BM microenvironment of all patients compared to normal BM. No difference was found between patient groups. Tissue factor (TF) expression was undetectable. The authors suggested that prothrombotic alterations are present in the BM microenvironment in patients with PCD, but also in the precursor state of MGUS [55]. The clinical relevance of these results remains to be elucidated.

### 2.3. Thrombogenic Properties of the M-Protein Predisposing to Arterial, Venous, or Microthrombosis

#### 2.3.1. Cryoproteins

i.Cryoglobulinemia

In type I cryoglobulinemia (CG) the cryoglobulin is a monoclonal immunoglobulin (Ig), usually IgM or IgG, and rarely IgA or free immunoglobulin light chain [11], that precipitates in vitro at cold temperatures and dissolves at 37 °C. Cryoglobulinemia (CG) is a laboratory finding, not necessarily associated with end-organ damage or a clinical syndrome. MGUS is seen in 40% of patients and the remaining have MM, chronic lymphocytic leukemia (CLL) or WM. In type II CG, mixed cryoglobulinemia (MC), a monoclonal Ig with rheumatoid factor activity binds to a polyclonal Ig heavy chain, which is in turn bound to an antigen. It is associated with HCV infection but hematological disorders, including B-lymphoproliferative disorders and MGUS, are also reported. The process of cryoprecipitation is not entirely understood; CGs become insoluble in cold temperatures, precipitate and form aggregates in small vessels causing obstructive microangiopathy in the microcirculation and thrombosis. Factors such as pH, ionic strength, CG concentration or calcium and chloride concentrations are also implicated in their aggregation. The phenomenon of precipitation is the result of a change in the balance between favorable/unfavorable interactions between CG and solvents at low temperatures. Altered glycosylation may be involved; immune stimulation may perhaps induce circulating desialylated immunoglobulins produced by B lymphocytes [56].

Disease presentation is variable and ranges from a mild clinical syndrome of fatigue, arthralgia and purpura to life-threatening complications. There is vascular occlusion by the cryoprecipitate, but small vessel vasculitis (purpura, glomerulonephritis and neuropathy) may also be seen (cryoglobulinemic vasculitis). Cutaneous manifestations (70–90%) are triggered by cold; acrocyanosis, cold-induced purpura and Raynaud phenomenon (RP) can be complicated with livedo reticularis, ulceration and necrosis [57,58]. Neurological symptoms (60–70%) and arthralgias (28%) are also reported. Renal disease (30% overall) is a late manifestation. Renal biopsy evidences membranoproliferative glomerulonephritis with glomerural thrombi, immune cell infiltration and microtubular deposits of cryoglobulin aggregates [58,59]. Thrombotic microangiopathy (TMA), mesenteric vasculitis, coronary vasculitis and hyperviscosity syndrome are also reported.

Treatment is indicated for the underlying symptomatic hematological disorder. When there is no indication for treatment (asymptomatic CLL or aWM) or the clonal cell expansion is very low, as seen in MGUS, the decision to initiate treatment should be made based on symptom severity. Recommendations are expert opinion-based, as the available data are scarce. Options for treatment include corticosteroids, alkylating agents, immunomodulatory agents, bortezomib and very often rituximab when the CG is IgM [60,61,62]. Symptom improvement is seen in 64–74% of patients when treatment is initiated [57,59].

ii.Cryofibrinogenemia

Cryofibrinogen is another type of cryoprecipitable protein. Unlike cryoglobulin, cryofibrinogen precipitates only in plasma. To detect cryofibrinogen in the laboratory, initial centrifugation at warm temperatures and subsequent cooling to 4 °C is required. A macroscopically visible precipitate is formed which resolubilizes upon warming [63].

Cryofibrinogenemia (CF) is a rare entity, and it is seldomly associated with a clinically relevant syndrome, although its rarity could be partly attributed to the technical difficulties in demonstrating the presence of the paraprotein. Isolated CF can be detected in up to 3% of healthy individuals and unselected patients admitted to hospital who are usually asymptomatic, but it may be associated with a higher incidence of thrombotic complications [64]. Plugging of deep and superficial blood vessels with thrombi that contain precipitates of cryofibrinogen, fibronectin, a1-antitrypsin and a2-macroglobulin are seen typically on skin biopsies [65]. A number of pathophysiological processes are potentially involved, including inhibition of fibrinolysis and altered thrombin-binding capacity. Manifestations of CF are similar to that of cryoglobulinemia and include cutaneous manifestations, Raynaud’s syndrome, cold sensitivity, arthralgia, renal disease and mononeuritis multiplex. Thrombotic complications are more common than in CG, reported in 5–56% of patients [64,65,66].

Available evidence suggests that CF and CG may co-exist, and the clinical significance of this association is largely unknown. The treatment approach in CF may include corticosteroids, immunosuppressants such as alkylating agents, therapeutic plasmapheresis, anticoagulation, aspirin or fibrinolytic agents with various degrees of success [66,67].

When fibrinogen complexes with a monoclonal antifibrinogenic antibody, we refer to the entity as type II cryofibrinogenemia. These cases are very rare; among 60 patients with CF reported in 2009, 24 had secondary cryofibrinogenemia and none of these cases were associated with underlying monoclonal paraprotein [66].

Euler et al., in 1996, reported the case of a patient who presented with cold-induced vasculitic purpura of the lower limbs, and arthralgia and investigation revealed the presence of a cryoprecipitate formed from fibrinogen plus an IgG3κ antibody. No monoclonal paraprotein was detected in standard serum immunoelectrophoresis (the paraprotein complexes with fibrinogen and is absent from serum). Affinity chromatography separation and subsequent immunofixation allowed detection of the paraprotein [68]. Control of symptoms was achieved by long-term plasmapheresis.

Cases of concurrent cryoglobulinemic vasculitis and cryofibrinogenemia that involves a monoclonal paraprotein are also very rare. In one case reported, a patient presented with cold-aggravated purpura vasculitis and leg ulceration. Monoclonal IgMκ and free κ was detected by serum electrophoresis, and immunophenotyping of the peripheral blood was consistent with low grade Β-cell lymphoproliferative disorder. The patient was treated with chlorambucil and plasmapheresis, aspirin and pentoxyphyline.

iii.Cryo-crystaloglobulinemia

Cryo-crystaloglobulinemia is a very rare complication of MG, first described in 1938 in the context of MM. A few reports have since been published. Most cases (approximately 50) have been associated with MM, but its association with non-MM MG has been established in recent years. There is spontaneous, reversible crystallization of monoclonal immunoglobulins (heavy and light chains) below 37 °C causing small vessel occlusion in the renal but also systemic vasculature, leading to ischemia and end-organ damage [69]. Kappa is more often the restricted light chain [70,71]. The clinical presentation has features of both type I and II cryoglobulinemia such as cutaneous purpura and ulcers, mucosal ulcers, erosive polyarthropathy, peripheral neuropathy, peripheral ischemia and rarely visceral ischemia and renal dysfunction. The renal lesion associated is referred to as crystalglobulin-associated nephropathy (CAN) [70,72]. Extracellular crystals of monoclonal paraproteins form pseudothrombi that occlude the renal glomerular capillaries and/or interstitial arterioles. The diagnosis is challenging as immunofluorescence is frequently negative and paraffin IF with pronase digestion is required. The diagnosis is usually based on the identification of intraluminal light chain crystals on electron microscopy. The therapeutic approach usually includes plasma exchange and chemo-immunosuppressive therapy directed against the underlying clone [73,74].

Leflot el al. reported the case of a 69-year-old woman who developed skin necrosis, purpura and rapidly progressive renal disease, monoclonal IgGκ and crystalglobulinemia. Immunofluorescence was negative, but crystalline materials within endothelial cells and subendothelial microtubular deposits were seen in electron microscopy, consistent with CCG-induced renal disease. Hemodialysis and plasma exchange were initiated and treatment commenced with bortezomib, cyclophosphamide and dexamethasone. Skin lesions improved but the patient remained hemodialysis dependent [75]. Leung et al. report a case of bilateral renal artery thrombosis associated with cryo-crystaloglobulinemia in a patient with MGUS [71]. Renal biopsy revealed immunoglobulin pseudothrombi that were crystalline in shape and stained for IgG and kappa light chain. No definite plasma cell disorder was identified in bone marrow biopsy. He initially received corticosteroids and azathioprine. Two years later he relapsed, and reevaluation revealed monoclonal IgGk, crystalline structures on electron microscopy and bilateral renal artery occlusion. There was also co-localization of the monoclonal immunoglobulin thrombin with fibrinogen, in support of its hypothesized thrombogenic nature. A case of CAN in a patient with underlying B-cell lymphocytosis and IgA kappa and free kappa light chain in serum electrophoresis has also been reported [76].

iv.Cold agglutinin disease and cold agglutinin syndrome

CAD is an autoimmune hemolytic anemia (AIHA) with a monospecific direct antiglobulin test (DAT) strongly positive for C3d and a cold agglutinin titer ≥ 64 at 4 °C. Patients may have a B-cell clonal lymphoproliferative disorder (LPD), but no clinical evidence of malignancy. Cold agglutinin syndrome is a similar entity, but a syndrome occurring secondary to another clinical disease [77,78,79]. Cold agglutinins are autoantibodies, the majority are IgM, but IgA or IgG have been reported and they are more likely to have kappa than lambda light chains [80,81,82,83].

The majority of cold agglutinins are IgMκ monoclonal antibodies that originate from a low-grade lymphoproliferative disorder, most often IgM-MGUS or WM [78]. Monoclonal antibodies bind to the erythrocyte surface resulting in agglutination and complement-mediated hemolysis [84,85]. The risk of thromboembolic events (TE) is increased in patients with CAD. The relative risk was 3.1 in one of the largest studies compared to matched controls [86]. One retrospective 10-year analysis reported an increased overall risk of TE of 29.6% in patients with CAD, compared to 17.6% in matched controls (HR 1.94, 95% CI 1.64–2.3) [86]. In cohorts from Europe and North America, the risk is mostly increased for venous TE and is less marked or absent for arterial TEs. The release of free heme, due to hemolysis, leads to nitric oxide scavenging and subsequently there is platelet aggregation, vasoconstriction and increased expression of endothelial adhesion molecules [87]. Another effect of the free heme is the generation of reactive oxygen species; cytokine release, inflammation and leukocytosis are enhanced [88,89]. Cytokine release in turn activates platelets and there is exposure of the subendothelial matrix due to endothelial injury [89]. Endothelial cell activation leads to tissue factor expression and macrovesicle release [90].

Indications for treatment initiation in CAD include severe anemia that requires transfusion or moderate/severe symptoms. Recurrent thrombotic events could also be an indication, but cases in the literature are scarce. Current treatment options include B-cell targeting and inhibition of antibody production or complement inhibition to prevent hemolysis. There are no randomized trials and no formal approval of any chemoimmunotherapy for CAD [91]. Another option seems to be classical complement pathway inhibition [92,93].Treatment for CAD reduces the frequency of TEs [93,94], but there are no data regarding the effect of treatment on the rate of TE in patients that have CAD but are asymptomatic.

#### 2.3.2. Autoantibody

i.Antiphospholipid (aPL) autoantibodies

The ability of monoclonal paraprotein to function as an autoantibody has been increasingly recognized and it has been linked to a number of systemic clinical presentations. Antiphospholipid (aPL) antibodies have been reported in the presence of monoclonal gammopathies [95,96,97]. These may have no clinical manifestation, or can be linked to thrombotic events. Monoclonal paraproteins with aPL-antibody activity have been described in patients with MM, WM and MGUS [98,99].

Among 93 patients with MGUS, a significantly higher incidence of aPL antibodies was reported compared to healthy controls [100]. Doyle et al. reported recently nine cases of antiphospholipid syndrome (APS) associated with MG [10]. The monoclonal immunoglobulin subtype was the same as the immunoglobulin subtype of the identified anticardiolipin and/or anti-β2-glycoprotein-1 antibody in all patients. The rate of thrombosis recurrence was 89% in patients with APS and monoclonal gammopathy vs. 42% in patients with no MG. In 67% of patients with APS and MG, the thrombotic event reoccurred during anticoagulation. Two patients had SMM, five MGUS and two lymphoplasmacytic lymphoma (LPL). Compared to patients with APS and no MG, the rate of recurrent thrombotic events was higher.

Monoclonal gammopathy and APS are independently associated with an increased thrombotic risk and the observed thrombotic events described might be due to the cumulative effect of the two. However, it is likely that the monoclonal paraprotein has aPL-antibody activity driving the prothrombotic environment. Isolation of the identified paraprotein to demonstrate its aPL activity is required.

There are also reports of lupus anticoagulant (LA)-like activity by monoclonal immunoglobulins which has been linked to thrombotic events [101].

#### 2.3.3. Disorders of Coagulation Pathway Inhibitors

Acquired activated protein C resistance has been reported in patients with MM in association with an increased risk of thrombosis [102,103,104]. There are no such reports yet in patients with non-symptomatic MG, but similar mechanisms could be responsible. MG has also been linked to the presence of acquired coagulation inhibitors in rare cases [105,106]. Antibodies against antithrombin, protein C and protein S have been reported [107,108].

ii.MG-induced thrombotic microangiopathy

Thrombotic microangiopathy (TMA) is a heterogenous clinical syndrome characterized by endothelial injury of variable etiology resulting in end-organ damage. In TMA, peripheral manifestations such as thrombocytopenia and microangiopathic hemolytic anemia (MAHA) are observed and also a renal syndrome which includes vascular thrombosis, mesangiolysis and subendothelial accumulation of cellular material [109]. There is growing evidence supporting the relationship between TMA and MG.

In one series, among 146 patients with TMA, MG was detected in 13.7%, with the most common being MGUS, followed by POEMS (polyneuropathy, organomegaly, endocrinopathy, monoclonal gammopathy, and skin changes) [110]. The authors suggest that MG in patients with TMA is approximately five times the expected frequency based on the incidence in the general population. It is, however, important to ensure that other TMA-related confounders such as infection, chemotherapy etc. are ruled out, in order to directly attribute TMA to the underlying MG. Yiu et al. reported a carefully selected cohort of nine patients with MG and TMA; five had MM, one WM and three MGUS [111]. All patients had renal TMA but not peripheral TMA.

It is possible that monoclonal proteins mimic the polyclonal immunoglobulins commonly involved in the pathogenesis of thrombotic thrombocytopenic purpura (TTP) or atypical hemolytic uremic syndrome (aHUS). Possible mechanisms include ADAMTS13 inhibitors, complement component (anti-factor H antibody) and interactions between monoclonal immnoglobulin and vWf or platelet membrane glycoprotein 1b [112,113]. Another presumed mechanism is disordered complement regulation, caused indirectly by the M-protein. The underlying mechanism of MG-associated TMA requires further study, but symptom improvement following treatment of the underlying plasma cell dyscrasia suggests an association [111]. Another important point is that MG induced TMA is commonly mostly limited to the kidney, with absence of peripheral TMA. Filippone et al. reported three cases of patients with kidney limited TMA. Two of the patients received clone-directed therapy and none received eculizumab [114]. TMA was therefore recently given a provisional status as one of the kidney lesions associated with monoclonal gammopathy of renal significance (MGRS) [72].

MGRS is a term used to describe kidney lesions associated with the presence of MG in the absence of other criteria for symptomatic disease. MGRS includes lesions due to direct paraprotein deposition or lesions due to the indirect activation of other mechanisms. (Cryo)crystaglobulin-associated nephropathy, cryoglobulinemic glomerulonephritis (type I and II CG) and cold agglutinin disease (CAD) are well characterized MGRS lesions [1]. In type I cryoglobulinemia and (Cryo)crystaglobulin-associated nephropathy, the disorder occurs due to deposition of the monoclonal immunoglobulin. In type II CG and CAD, the monoclonal immunoglobulin has autoantibody activity that mediates the renal lesion.

iii.Other rare entities

Greinacher et al. recently reported the case of an unvaccinated patient with monoclonal gammopathy with a state of chronic hypercoagulability mimicking vaccine-induced immune thrombocytopenia (VITT) and characterized by the presence of platelet-activating anti-PF4 antibodies, not associated with heparin treatment. This case points further to the multifaceted character of the monoclonal paraprotein and its role as the underlying cause of chronic prothrombotic disorders [115].

## 3. Discussion

Several considerations should be taken into account when interpretating data on MGUS-related thrombotic risk. MGUS is usually an asymptomatic condition, and its diagnosis is typically incidental during clinical workup for unrelated medical issues. However, in some patients, it is associated with clinically significant features such as infections, fractures, neuropathy and increased mortality. It should therefore be acknowledged that the MGUS population is heterogeneous, and this is reflected in both clinical and biomarker studies assessing the magnitude of thrombotic risk. Historical studies involved patients referred to a hospital, either for MGUS or VTE or both. However, in hospital-based studies, disease associations may occur coincidentally considering that M-protein testing is performed more frequently in patients with certain clinical presentations. Moreover, clustering of VTE around the time of MGUS diagnosis is consistent with a protopathic bias, while a more constant risk over time would be expected if thrombosis risk was an innate feature of MGUS. MGUS cohorts therefore may not be representative of the general MGUS population, introducing selection bias. Population-based studies appear to confer contraindicatory results in terms of the associated thrombotic risk. Nevertheless, what seems to be consistent is that screened MGUS and incidentally diagnosed MGUS are two distinct groups, with the latter presenting probably a higher thrombotic risk. This could be partly explained by the presence of classical thrombotic risk factors (reduced immobility, more frequent hospitalizations, comorbidities). Whether this is also a question of a differing coagulation profile in these two populations has never been addressed in biomarkers studies. Further studies are required to better understand the observed differences between studies and across populations. Future studies should distinguish MGUS patients based on their comorbidities when compared to matched populations without M-protein.

Concerning evidence from biomarker studies, hypercoagulability and endothelial activation are consistent findings in MGUS patients. Platelets with a hyporeactive phenotype is also a common observation. Baseline platelet hyperactivation, presumably related to ongoing plasma cell activity, could conceivably result in hypo-responsiveness, which has been shown to correlate with poor prognosis in MM patients. However, clinical relevance of these findings in MGUS patients remains largely unknown. Moreover, group heterogeneity and the small number of patients included represent major drawbacks of these studies. At the moment, data are insufficient to draw safe conclusions. Novel markers, such as circulating DNA and translational studies of the BM environment, are promising and could elucidate some aspects of the multifactorial etiology of MG-related thrombosis.

Absolute numbers of VTE are too low to justify use of systematic thromboprophylaxis in patients with MGUS. Rather, attention should be drawn to coexisting medical conditions in MGUS patients that warrant thromboprophylaxis (hospitalization, immobilization, surgical interventions or long-haul flights). However, considering MGUS as a minor but persistent VTE risk factor could justify extended anticoagulation in VTE patients, especially in the presence of other known VTE risk factors. Furthermore, serum protein electrophoresis should be included in the work-up of patients with extensive and unprovoked venous thromboembolic events. A difficult question to answer is whether there is a subset of patients presenting with thrombotic events that warrants treatment initiation targeting the underlying, usually small, plasma cell clone. Patients with established MG and unexplained, recurrent thrombotic events despite anticoagulation, events in both the arterial and venous beds, or including the microcirculation with systemic manifestations with eminent end-organ damage, should prompt the clinician to consider a link between M-protein and the thrombotic manifestations. In some patients, establishing causality between MG and thrombosis is feasible (coexistence of APL antibodies, positive anti PF4 antibodies, antibodies against coagulation inhibitors, presence of cryoproteins and kidney or skin biopsy). 

Bearing in mind the large disease burden of the thromboembolic disease and based on the data presented in this review, it becomes evident that in a subset of patients, the M-protein may acquire a role that has “thrombotic significance”. Thus, we propose that further investigation is warranted, in order to define a potential umbrella term of “monoclonal gammopathy of thrombotic significance” (MGTS). This is a provisional term, aiming to ignite discussions and investigations in the field and not a definite syndromic diagnosis. However, since the adoption of such a term may have clinical implications that no longer justify a monitoring approach and should potentially prompt appropriate intervention and treatment, there needs to be substantial research and prospective evaluation. Today, with the currently available data, is not possible to provide precise criteria for this umbrella term and as data continues to accumulate, the entity needs to become more well-defined. Thus, the provisional term “MGTS” could be considered as a syndromic diagnosis on an individualized basis when all the following are met: (1) a monoclonal paraprotein is present, (2) the criteria for multiple myeloma and other plasma cell dyscrasias that warrant treatment are not met, and (3) there are thrombotic events or complications that cannot be otherwise be explained and have significant clinical implications for the patients. When all these criteria are met, then considerations for long-term anticoagulation and, in some life-threatening circumstances, anti-clonal therapy should be discussed and weighed against potential risks.

## 4. Conclusions

What is undoubtedly clear, based on currently available evidence, is that the thrombotic risk in patients with MGUS shows significant heterogeneity. A subset of MGUS patients, who are otherwise asymptomatic and do not meet criteria for treatment initiation, seem to have clinically significant thrombotic manifestations. The M-protein can have distinct prothrombotic properties like that of a cryoprotein or of an antibody with antiphospholipid activity, and could be the missing link in these patients, particularly in a context of unexplained, recurrent, or even catastrophic thrombotic events. We propose a new term for these entities, monoclonal gammopathy of thrombotic significance, which expands further the list of syndromes that fall under the umbrella term monoclonal gammopathy of clinical significance. The challenge remains to distinguish the high-risk patients for this entity among MGUS patients, using biomarker fingerprinting along with clinical profiling. Diagnosis should be considered after ruling out other common conditions related to thrombosis and requires establishing a link between thrombotic events and M-protein. The presence of thrombotic events with severe presentation, recurrent character, in unusual sites, in both arterial and venous vascular beds, or associated to thrombosis in microcirculation (skin, renal thrombosis) should raise suspicion for syndromes that meet the criteria for monoclonal gammopathy of clinical significance and should pull the trigger for testing for cryoproteins, APL antibodies and autoimmune HIT. Patients with monoclonal gammopathy of thrombotic significance should be candidates for treatment initiation directed against the plasma cell clone to eliminate the monoclonal protein, which initiates the processes that eventually result in thrombotic events. 

## Figures and Tables

**Table 1 cancers-15-00480-t001:** Main population-based peer-reviewed studies reporting the incidence of venous thromboembolism and associated independent risk factors confirmed in multivariate analysis) in patients with Monoclonal gammopathy of undetermined significance (MGUS). MM: multiple myeloma, PCD: plasma cell dyscrasias, LPD: lymphoproliferative disorder, HC: healthy controls, AL: AL amyloidosis, pts: patients, VTE: venous thromboembolism, DVT: deep venous thromboembolism, X: not reported, CVD: cardiovascular disease, M-protein: monoclonal protein, HR: hazard ratio, WBC: white blood cells. IRR: Incidence rate ratio, PY: person-years, * Confirmed by multivariate analysis.

	Type of Study	Population	Outcomes	Independent * Risk Factors for Thrombosis	Strengths	Weaknesses
1	Single center Retrospective Hospital based Srkalovic G, 2004 [15]	404 MM 174 MGUS 34 other (non-AL)	VTE in 7.5% MGUS and 10% in MM pts (increased compared to VTE rates in the general population)	Family VTE Hx (HR = 13.79; 95% CI, 1.69–112.8) Personal VTE Hx (HR = 8.95; 95% CI 2.28–35.10) Immobility (HR = 27.71 (95% CI; 5.41–141.9) Low albumin levels (HR = 4.16 95% CI; 1.75–10) High WBC (HR = 1.2 95% CI; 1.08–1.33)	Detailed medical history Clinical information on site of VTE events	Retrospective No control group
2	Single center Prospective Hospital based Sallah S, 2004 [16]	310 MGUS - 269 outpatients - 41 hospitalized	VTE in 6.1% MGUS (increased compared to VTE rates in the general population)	Serum M-protein > 16 g/L (RR 6.3; 95% CI 2.25–17.6) Progression to PCD and LPD (RR 4.2; 95% CI 1.64–10.7)	Prospective Additional information on VTE events	No control group
3	Inpatients in veteran’s hospitals Kristinsson et al., 2008 [17]	2374 MGUS and 6192 MM among 4,196,197 Veterans	DVT incidence: 3.1 per 1000 PY in MGUS, 8.7 per 1000 PY in MM RR 3.3-fold (95% CI, 2.3–4.7) in MGUS pts, 9.2 (95% CI, 7.9–10.8) in MM pts	x	Long follow-up Large cohort African American and white veterans included	No data on comorbidities Retrospective Restricted to hospitalized males Diagnosis made with the discharge diagnostic codes Isolated PE not identified
4	Population-based Data from Swedish Cancer Registry and nationwide MGUS cohort Kristinsson et al., 2010 [18]	18,627 MM pts vs. 70,991 HCs 5326 MGUS pts vs. 20,161 HCs	At 1, 5, and 10 years after MGUS diagnosis: HR for VTE 3.4 (2.5–4.6), 2.1 (1.7–2.5), 2.1 (1.8–2.4) HR for arterial thrombosis: 1.7 (1.5–1.9), 1.3 (1.2–1.4), 1.3 (1.3–1.4)	IgG/IgA M-protein: HR at 1-year follow-up - 4.2; 95% CI, 2.6–6.8 for VTE - 1.6; 95% CI, 1.3–1.9 for arterial thrombosis	Large size Register-based cohort Most MGUS pts underwent a bone marrow examination Arterial events recorded	Lack of detailed clinical data Lack of information on potential confounders
5	Hospital-based Controlled Single-center Prospective Cohen et al., 2010 [19]	166 MGUS 465 non MGUS pts	VTE incidence in MGUS pts similar to non- MGUS pts, HR 1.38 (95% CI; 0.63–3.01, NS) Increased compared to VTE rates in the general population	Personal history of VTE (HR 3.33; 95% CI 1.26–8.8) Albumin level (HR 0.22; 95% CI 0.1–0.45)	Prospective Complete medical records	Only male hospitalized veterans Over half of the pts > 70 years old Not all MGUS pts had complete work-ups for MM
6	Single center Retrospective Hospital based Muslimani et al., 2009 [20]	112 MGUS pts	VTE incidence: 8% (increased compared to VTE rates in the general population)	x	Exclusion of pts with VTE risk factors	Retrospective Small number No control group Short follow-up
7	Population-based Screening Bida et al., 2009 [21]	605 MGUS 16,793 controls	Sural phlebitis (HR = 8.8; 95% CI, 2–27.3) Popliteal artery embolism (HR = 7.8; 95% CI, 2–30.7) Femoral artery embolism (HR = 7.1; 95% CI, 1.9–26.6)	x	Screened population	Disease definition based on diagnostic codes
8	Population-based Danish National Patient Registry Gregersen et al., 2011 [22]	1610 MGUS pts 16,100 control subjects	4.0 vs. 3 VTEs/1000 PY IRR 1.37; 95% CI, 1.00–1.88	x	Large cohort Population comparison cohort with no VTE Hx and matched by comorbidities	Coding errors Control subjects not screened for M-protein
9	Population-based Multicenter Retrospective Za et al., 2015 [23]	1491 MGUS pts (1238 with a follow up >12 months)	Thrombotic Incidence *: 4.49 per 1000 patient-years (2.59 for arterial events and 1.90 for venous events)	For VTE events =: M-protein > 16 g/L (HR 3.08, 95% CI 1.01–9.36). For arterial thrombosis: CVD risk factors (HR 4.92; 95% CI 1.42–17.04)	Population based Estimation of arterial events	Retrospective No control group

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
