# Peer review of "Monoclonal Gammopathy of Thrombotic Significance"

_cancers, 2023, doi:10.3390/cancers15020480_

Round 1
Reviewer 1 Report
The authors have described a comprehensive review of MGUS with thrombotic complications. The manuscript is overall well done; and I thank the authors for their excellent review.
Minor comments:
1. Under introduction, please remove ‘European,’ and modify the sentence per the cited reference#2. The prevalence of MGUS is estimated at 3.2% by Kyle RA et al. study in about 21,000 patients in Olmsted County.
2. Consolidate the discussion under the clinical trial data section into fewer paragraphs while keeping the context intact.
Author Response
The authors have described a comprehensive review of MGUS with thrombotic complications. The manuscript is overall well done; and I thank the authors for their excellent review.
Minor comments:
- Under introduction, please remove ‘European,’ and modify the sentence per the cited reference#2. The prevalence of MGUS is estimated at 3.2% by Kyle RA et al. study in about 21,000 patients in Olmsted County.
We have made the relevant correction, thank you very much for pointing this out.
- Consolidate the discussion under the clinical trial data section into fewer paragraphs while keeping the context intact.
Thank you for your input. We have shortened the relevant section.
Reviewer 2 Report
In this paper, Vasiliki Gkalea et al., made a review based in MGUS, MM and other rare conditions that associated MG with the thrombogenic properties of the monoclonal protein. They want to demonstrate that the M-protein can be linked to clinically significant thrombotic events. Based on that clinical studies the authors propose a new term “monoclonal gammopathy of thrombotic significance” (MGTS) to refer to significant, recurrent thrombotic events in patients with MGUS.
Besides, the subject of the article being very interesting and original, the proposed new term MGTS is not well founded. The article is limited to the description of several studies, some even contradictory, but in the end there is no concrete proposal that defines the new entity proposed monoclonal gammopathy of thrombotic significance. The author must correlate all the studies in order to define a profile that defines the MGTS clearly. A meta analysis might make the article more robust.
Author Response
In this paper, Vasiliki Gkalea et al., made a review based in MGUS, MM and other rare conditions that associated MG with the thrombogenic properties of the monoclonal protein. They want to demonstrate that the M-protein can be linked to clinically significant thrombotic events. Based on that clinical studies the authors propose a new term “monoclonal gammopathy of thrombotic significance” (MGTS) to refer to significant, recurrent thrombotic events in patients with MGUS.
Besides, the subject of the article being very interesting and original, the proposed new term MGTS is not well founded. The article is limited to the description of several studies, some even contradictory, but in the end there is no concrete proposal that defines the new entity proposed monoclonal gammopathy of thrombotic significance. The author must correlate all the studies in order to define a profile that defines the MGTS clearly. A meta analysis might make the article more robust
Dear reviewer thank you for your input and consideration of our work. We agree that a meta-analysis could provide a more in depth evaluation of the correlation of thrombotic events with monoclonal gammopathies, however, this is out of the scope of this review and second the heterogeneity of the studies makes the results of such an effort difficult to interpret. The term MGTS needs further validation and the purpose of this review is to provide feedback and ignite discussion and further investigation in the field. We emphasize that this is a provisional term that clearly needs further investigation and prospective evaluation, given the potential clinical impact of such terminology. In this regard we have modified the relevant discussion as follows: “Bearing in mind the large disease burden of the thromboembolic disease and based on the data presented in this review, it becomes evident that in a subset of patients, the M-protein may acquire a role that has “thrombotic significance”. Thus, we propose that further investigation is warranted in order to identify a potential umbrella term of “monoclonal gammopathy of thrombotic significance” (MGTS). This is a provisional term, aiming to ignite discussions and investigations in the field and not a definite syndromic diagnosis. However, since the adoption of such term may have clinical implications that no longer justify the clinician to monitor the patient and may prompt appropriate intervention and treatment, there needs to be substantial research and prospective evaluation. Today, with the currently available data, is not possible to provide precise criteria for this umbrella term, as data continues to accumulate the entity needs to become more well-defined. Thus, the provisional term “MGTS” could be considered as a syndromic diagnosis on an individualized basis when all the following are met: (1) a monoclonal paraprotein is present (2) the criteria for multiple myeloma and other plasma cell dyscrasias that warrant treatment are not met and (3) there are thrombotic events or complications that cannot be otherwise be explained and have significant clinical implications for the patients. When all these criteria are met, then considerations for long-term anticoagulation and in some life-threatening circumstances anti-clonal therapy should be discussed and weighed against potential risks.”
Reviewer 3 Report
In their manuscript the authors provided a comprehensive review on the association between monoclonal gammopathy and thrombotic events. Data derived from the clinical trials as well from biomarker studies are well discussed and a future perspective concerning therapeutical approach is given.
Due to the fact that both monoclonal gammopathy and multiple myeloma may also cause bleeding disorders (such as acquired von Willebrand disease) we suggest to include in the Introduction a short overview on the haemostatic disturbances in plasma cell dyscrasias.
Author Response
In their manuscript the authors provided a comprehensive review on the association between monoclonal gammopathy and thrombotic events. Data derived from the clinical trials as well from biomarker studies are well discussed and a future perspective concerning therapeutical approach is given.
Due to the fact that both monoclonal gammopathy and multiple myeloma may also cause bleeding disorders (such as acquired von Willebrand disease) we suggest to include in the Introduction a short overview on the haemostatic disturbances in plasma cell dyscrasias.
Thank you for your comments and input. We have added the following paragraph in the introduction: The complex pathophysiologic interactions between M-protein, platelets, coagulation factors and endothelium, can also produce hemostatic abnormalities that increase the bleeding risk.14 It is beyond the scope of this review to discuss MG - related bleeding disorders such as acquired von Willebrand syndrome, FX deficiency and platelet function disorders. . However, the notion that M-protein can cause both thrombotic and hemorrhagic tendencies emphasizes further its unique physicochemical properties.
Round 2
Reviewer 2 Report
The authors had answer almost all the previous questions and concerns.